# Biotechnological Strategies for Chitosan Production by Mucoralean Strains and Dimorphism Using Renewable Substrates

**DOI:** 10.3390/ijms21124286

**Published:** 2020-06-16

**Authors:** Adriana Ferreira de Souza, Hugo Marques Galindo, Marcos Antônio Barbosa de Lima, Daylin Rubio Ribeaux, Dayana Montero Rodríguez, Rosileide Fontenele da Silva Andrade, Norma Buarque Gusmão, Galba Maria de Campos-Takaki

**Affiliations:** 1Biotechnology Post-Graduation Program, Federal Rural University of Pernambuco, Recife-PE 52171-900, Brazil; adrife.souza@gmail.com; 2Fungal Biology Post-Graduation Program, Federal University of Pernambuco, Recife-PE 50670-420, Brazil; hugo.galindo21@gmail.com; 3Department of Microbiology, Federal Rural University of Pernambuco, Recife-PE 50670-420, Brazil; mablima33@yahoo.com.br; 4Biological Sciences Post-Graduation Program, Federal University of Pernambuco, Recife-PE 50670-420, Brazil; drubioribeaux@gmail.com; 5Post-Doctorate National Program-CAPES, Catholic University of Pernambuco, Recife-PE 50050-900, Brazil; dayanamontero87@gmail.com (D.M.R.); rosileide_fontenele@yahoo.com.br (R.F.d.S.A.); 6Department of Antibiotics, Federal University of Pernambuco, Recife-PE 50670-901, Brazil; normagusmao@gmail.com; 7Nucleus of Research in Environmental Sciences and Biotechnology, Catholic University of Pernambuco, Pernambuco 50050-590, Brazil

**Keywords:** agro-industrial wastes, biopolymer, Mucorales, morphology, pellets

## Abstract

We investigated the influence of corn steep liquor (CSL) and cassava waste water (CWW) as carbon and nitrogen sources on the morphology and production of biomass and chitosan by *Mucor subtilissimus* UCP 1262 and *Lichtheimia hyalospora* UCP 1266. The highest biomass yields of 4.832 g/L (*M. subtilissimus* UCP 1262) and 6.345 g/L (*L. hyalospora* UCP 1266) were produced in assay 2 (6% CSL and 4% CWW), factorial design 2^2^, and also favored higher chitosan production (32.471 mg/g) for *M. subtilissimus*. The highest chitosan production (44.91 mg/g) by *L. hyalospora* (UCP 1266) was obtained at the central point (4% of CWW and 6% of CSL). The statistical analysis, the higher concentration of CSL, and lower concentration of CWW significantly contributed to the growth of the strains. The FTIR bands confirmed the deacetylation degree of 80.29% and 83.61% of the chitosan produced by *M. subtilissimus* (UCP 1262) and *L. hyalospora* (UCP 1266), respectively. *M. subtilissimus* (UCP 1262) showed dimorphism in assay 4–6% CSL and 8% CWW and central point. *L. hyalospora* (UCP 1266) was optimized using a central composite rotational design, and the highest yield of chitosan (63.18 mg/g) was obtained in medium containing 8.82% CSL and 7% CWW. The experimental data suggest that the use of CSL and CWW is a promising association to chitosan production.

## 1. Introduction

In recent times, chitosan is one of the most important biopolymers on the market, due to its properties such as biocompatibility, biodegradability, adsorption ability, and antimicrobial activity, with wide applications in several areas. However, special attention has been paid to the development of new processes and sustainable methods of synthesis of this polysaccharide [1].

The chitosan is a heterogeneous polysaccharide containing 2-amino-2-deoxy-D-glucose linked by β 1-4 bonds (glycosamine) and *N*-acetyl-2-amino-2-deoxy-D-glucose (*N*-acetyl-glycosamine), consisting of more than 60% of glycosamine [1,2]. Its cationic nature, chelating ion-binding capacity, protein immobilization, film and gel formation characteristics, chemical reactivity, modifiability, biocompatibility and antimicrobial activity are all suggested due to their heterogeneous structure and the presence of protonatable amino group of d-glycosamine [3,4,5].

Commercial chitosan is derived from the deacetylation of chitin from crustacean residues, where the process requires a strong alkaline treatment with high temperature due to the complex organization of chitin with other components in the exoskeleton crustacean, and is a slow process with a high cost, causing environmental pollution due to chemical treatments [6]. In addition, this raw material is seasonally and geographically limited and has highly variable characteristics, which may interfere with product yield and molecular product characteristics [5].

Filamentous fungi, mainly Mucoromycota, formerly belonging to a polyphiletic phylum Zigomycota [7], are known to synthesize natural chitosan in their cell wall, produced by enzymatic deacetylation of the chitin chain, a modification of sugar chains [6,8,9,10,11,12,13,14].

The production of chitosan from the fungal cell walls presents advantages over the traditional process for deacetylation of shrimp residues. The bioprocess of chitosan extraction does not require high temperatures and strong alkaline solutions using simple chemical procedures, and moreover has a low fungal biomass content which compared with inorganic materials compared to crustacean wastes. The chemical process used to crustacean require the demineralization as well as heavy metals such as nickel and copper contamination the environment. The chitosan-producing fungal species have approximately the same deacetylation of degree and lower viscosity than chitosan obtained from the deacetylation chitin from crustaceans [5,6,11,15]. In addition, fungal chitosan is produced in a controlled and seasonally independent environment [1,5,16].

The fungal cell wall suggested sustainable route for chitosan production using agroindustrial residues. This article evaluated chitosan production by *Mucor subtilissimus* UCP 1262 and *Lichtheimia hyalospora* (UCP 1266) cultivated by submerged fermentation using agroindustrial residues, cassava waste water, and corn steep liquor; alternative sources; as well as the effect on dimorphism. In addition, the chitosan produced were characterized in terms of their acetylation degree and viscosity, and their properties were compared with those of a commercial chitosan.

The microbiological bioprocess investigation of chitosan production by fungi was based on problems with commercial sources from shrimp and crabshell chitin deacetylated by strong alkali solution at high temperatures for long periods of time, being a source of seasonal and limited supply, as well as causing environmental pollution if inadequately discarded. Thus, it is of fundamental importance that Mucorales fungi order be used as natural chitin and chitosan sources, which is produced in their cell wall during their growth, for the purpose of the development, innovation, and generation of green technology using agro-industrial substrates, contributing to the reduction of these residues in the environment. The production of a bioactive biopolymer may partially satisfy a part of the need for innovative products in the biomedicine, pharmaceutical, and environmental areas, as it is a natural resource and has low toxicity, moreover guaranteeing positive impacts for the economy, as well as the possibility of a new process being applied to industry in the future.

## 2. Results and Discussion

### 2.1. Elemental Analysis of Cassava Waste Water (CWW) and Corn Steep Liquor (CSL)

The carbon, nitrogen, oxygen, and sulfur concentrations of CWW and CSL are shown in Table 1. CSL is the main source of carbon and nitrogen and with CWW provide essential nutrients for the growth of microorganisms.

### 2.2. Effect of Substrate Concentrations on Fungal Morphology

In this study, the effect of the concentrations of corn steep liquor (CSL) and cassava waste water (CWW) used in 2^2^ factorial design (Table 2) on fungal morphology was evaluated. Microscopic and macroscopic images showed cellular differentiation in *M. subtilissimus* (UCP 1262) (Figure 1) and *L. hyalospora* (UCP 1266) (Figure 2).

For *M. subtilissimus* (UCP 1262), the yeast phase was evident in assays 3 (Figure 1c) and 4 (Figure 1d) (higher concentrations of CWW, 8%). In the remaining assays, the transition phase (presence of globose cells, longs/shorts, and arthroposporous hyphae) was observed (Figure 1). In the assays 1 (Figure 1a) and point central (Figure 1e), yeast-like cells and fragments of short arthroposporous hyphae were observed. In assay 2 (Figure 1b), we observed yeast cells and long arthroposporous hyphae. In addition to mycelium formation, different filamentous fungal species can grow in yeast, depending on environmental conditions. This transition is called dimorphism [17].

Karimi and Zamani [18] stated that in addition to factors and mechanisms such as initial spore concentration, sugar concentration, and atmospheric factors evaluated in several studies [18,19,20,21,22], fungal morphology may be affected by the addition of certain chemical compounds. Among them, cyanide, acriflavine, and cycloheximide induce yeast morphology due to the inhibition of synthesis or action of cytochrome oxidase. These results can be corroborated in the current study, considering that CWW presents high cyanide concentrations (444.0 mg of cyanide per liter of CWW) [23,24].

In submerged cultivation with constant agitation, filamentous fungi exhibit three macroscopic morphologies—scattered mycelia, clumps (small pellets), and pellets (spherical masses of hyphae) [25,26]. However, due to the predominance of the yeast and transition phase, for *M. subtilissimus* (UCP 1262), the granular aspect was observed under experimental conditions (Figure 1f–j). In terms of the predominance of granular mycelium, a few areas of differentiation/fragmentation were observed in assays 1 (Figure 1f) and 2 (Figure 1g). Assay 3 (Figure 1i) and center point (Figure 1j) presented granular mycelium with fragmentation areas. Presence of scattered mycelia with significant presence of areas of differentiation and viscous liquid were observed in assay 3 (Figure 1h).

Unlike *M. subtilissimus* (UCP 1262), *L. hyalospora* (UCP 1266) showed no dimorphism. No microscopically relevant differences in agro-industrial waste concentrations were observed (Figure 2). *L. hyalospora* (UCP 1266) produced different sizes and shapes of pellets that differed with agro-industrial waste concentrations. Hollow pellets over 2 mm in diameter were predominant in all assays (Figure 2f–j), and microscopically with hyphae without cell lysis (Figure 2a–e), except assay 3 (Figure 2h; highest CWW level and lower CSL level), with compact and smooth pellets ranging from 1 to 2 mm in diameter and the presence of fragmented hyphae (Figure 2c). Large and hollow pellets have been observed in several studies [12,27,28,29], where it has been suggested that autolysis in the center of these clusters of hyphae is caused by difficulties in oxygen transfer.

According to [25], the composition of the medium can influence the pellet structure, as high concentrations of nitrogen induce pellet formation. Our results differ from those obtained by [30], who showed that CSL did not influence pellet formation in *Penicillium chrysogenum*. However, Metz and Kossen [25] reported that the type of fungal strain is an important factor influencing pellet formation.

### 2.3. Effect of Substrates on Biomass and Chitosan Production by M. subtilissimus (UCP 1262) and L. hyalospora (UCP 1266)

Although both fungi showed different patterns in macro-morphology, both showed high potentials in the assimilation of alternative sources of carbon and nitrogen.

Table 3 presents the comparative analysis of the results obtained for biomass and chitosan production in each 2^2^ factorial design. The highest biomass yield produced by *M. subtilissimus* (UCP 1262; 4.832 mg/L) and *L. hyalospora* (UCP 1266; 6.540 mg/L) was in the culture medium supplemented with CSL 6% and CWW 4%, in which *M. subtilissimus* (UCP 1262) presented predominance of long and arthrosporous hyphae and *L. hyalospora* (UCP 1266) formed pellets of 4 to 5 mm in diameter. The result suggests that media with higher concentrations of CSL favor the growth of the fungi studied. Similar results were reported by Berger et al. (2014), when using the CSL and CWW in the culture medium for growth by *Cunninghamella elegans*. CSL is a carbohydrate- and amino acid-rich residue that favors the growth of filamentous and unicellular fungi [23,31,32].

An increase in pH from 6.0 (beginning of fermentation) to 7.1–6.2 was observed in *M. subtilissimus* (UCP 1262) cultivation, a result similar to that of [23,33,34]. Lower pH resulted in higher biomass yields, in line with that described by [23].

However, at the end of *L. hyalospora* (UCP 1266) cultivation there was a decrease in pH from 6.0 (initial pH) to 5.1. Similar results were observed by [35,36]. Just as low pH favors fungal growth, it also favors chitin deacetylase activity, increasing chitin deacetylation in chitosan [37,38].

As for biomass production, assay 2 provided the best chitosan yields by *M. subtilissimus* (UCP 1262) (32.47 mg/g), unlike [23], which obtained the best chitosan yield by *C. elegans* in higher concentration of CWW and lower concentration of CSL [39] when cultivating *C. elegans* in medium, with higher concentration of corn steep liquor—CSL (7%) obtaining the best chitosan yield. The study by [40], when cultivating *Syncephalastrum racemosum* in higher concentrations of CSL (8%), obtained the highest yield of biomass and chitosan.

Similar to *M. subtilissimus* (UCP 1262), assay 2 (6% CSL and 4% CWW) provided the highest biomass production (6.540 g/L) by *L. hyalospora* (UCP 1266). However, it obtained the highest chitosan production (45.03 mg/g) at the central point of the fractional factorial design-FFD (4% of CSL and 6% of CWW). Previously, *Rhizopus arrhizus* obtained a higher production of the biopolymer also using 4% of CSL [41].

Higher concentrations of cassava waste water negatively influenced the chitosan production by *M. subtilissimus* (UCP 1262)*,* as well as fungal morphology, predominantly forming individual spherical cells and budding multiplication. This yeast phase probably contributed negatively to chitosan production. This assumption is shown in Figure 1 in assay 3 (2% corn steep liquor and 8% cassava waste water) and assay 4 (6% of corn steep liquor and 8% cassava waste water). The influence of dimorphism on chitosan yield was also observed by [20], who detected in the cell wall of *M. rouxii* (now *Rhizopus arrhizus*) that the filament form had higher chitosan yields (9.4%) compared to the yeast phase, which had lower chitosan contents (8.4%).

The same condition that stimulated the *M. subtilissimus* (UCP 1262) yeast phase, resulting in low chitosan yields, influenced the formation of compact and smooth pellets by *L. hyalospora* (UCP 1266), also causing low biomass and chitosan yields. The best pellet diameter range for chitosan production was 4 to 5 mm, with a content of 45.03 g/kg of dry biomass. In the study of Sparringa et al. [42] the highest production of glycosamine (107 g/kg), obtained from chitosan deacetylated produced by *Rhizopus oligosporus* NRRL 2710, resulted in pellet formation with 16–35 mm. Sparringa and Owens [43] analyzed the pellet size of affected *R. oryzae* ATCC 20,344 glycosamine production. In addition, high glycosamine content (0.19 g/g) was obtained with pellet formation of 5.0 mm, and reduction to 1 mm negatively influenced glycosamine yield to 0.15 g/g.

Ultrastructural aspects of biomasses with higher chitosan yields were also observed by scanning electron microscopy. Figure 3 shows the mycelial branches of the *Mucor subtilissimus* (UCP 1262) (6% CSL/4% CWW) with loose mycelium and hyphae with a thickness that was thinner, tubular, contorted, and with morphological aspect of yeast cells (Figure 3a) and *Lichtheimia hyalospora* (UCP 1266) (assay central point: 4% CSL/6% CWW) with compact mycelium hyphae with a thickness that was thinner, tubular and contorted. (Figure 3b). This study suggests that *L. hyalospora* (UCP 1266) mycelium fragmentation is a result of pellet formation. As other genera of the order Mucorales, they grow rapidly and have tubular-shaped hyphae and no septation. Liao et al. [44] observed morphological aspects of *Rhizopus oryzae* by SEM grown in agroindustrial wastes (soybean meal, wheat, and rice) to understand the relationship between morphology and production of glycosamine, lipids, and amino acids. According to [45], the culture medium composition and culture conditions influence the metabolic regulation and, consequently, the morphology. The increase in biomass and chitosan is a reliable indicator of the development of the studied fungi.

Table 4 presents the best results from various studies for biomass and chitosan production by Mucoralean fungi, which suggests that the chitosan content of the fungi depends on the fermentation time, culture medium, and cultivation conditions.

Figure 4 presents the influence of the independent variables, corn steep liquor—CSL (1) and cassava waste water-CWW (2), and the interaction between these variables (1 × 2) on biomass production by *Mucor subtilissimus* (UCP 1262) and *Lichtheimia hyalospora* (UCP 1266), using factorial design, with statistical significance of *p* < 0.05. The Pareto chart illustrates that the increase in the concentration of CSL (1) positively influenced the growth of the microorganisms. However, lower levels of CWW (2) were suggested for biomass production (Figure 4a,b). The interaction of the independent variables (1 × 2) showed that the maximum level of CSL (1) and the minimum level of CWW (2) had an antagonistic interaction, with significant influence on the biomass production in both microorganisms (Figure 4a,b).

The Pareto graph (Figure 5a) proves that higher levels of the independent variable CSL (1) significantly influenced the chitosan production. However, lower levels of the CWW (2) significantly influenced the production of chitosan by *M. subtilissimus* (UCP 1262). The interaction of the independent variables (1 × 2) significantly influenced, antagonistically, the chitosan production by *M. subtilissimus*.

The Pareto (Figure 5b) shows that higher concentrations of both independent variables—CSL and CWW—exhibited a significant influence on the chitosan production by *L. hyalospora* (UCP 1266). Consequently, the interaction of the variables (1 × 2) was significant, suggesting that the synergic effect of these substrates stimulated the biopolymer production.

### 2.4. Characterization of Chitosan Extracted from M. subtilissimus (UCP 1262) and L. hyalospora (UCP 1266)

#### Degree of Deacetylation

Chitosan extracted from the fungal biomass of *M. subtilissimus* (UCP 1262) and *L. hyalospora* (UCP 1266) were characterized by FTIR spectroscopy. The data obtained allowed the identification of extracted chitosan as well as the estimation of the degree of deacetylation (DD%), a fundamental parameter that influences the biological and physicochemical properties of the biopolymer. The degree of deacetylation of the fungal chitosan is shown in Table 5 and Figure 6. The amide I and amine bands of chitosans extracted from *M. subtilissimus* (UCP 1262) were found at 1642-1545 cm^−1^. In the case chitosan obtained from *L. hyalospora* (UCP 1266), the analysis from the FTIR spectrum showed that the amide I peak was observed at 1649 cm^−1^, and the amine peak was 1556 cm^−1^. The values are consistent with those reported by the literature [8,9,10,23,31].

As is observable from Figure 6, the bands at 1545 cm^−1^ (*M. subtilissimus* UCP 1262) and 1556 cm^−1^ (*L. hyalospora* UCP 1266) showed significant intensities, suggesting stable deacetylation in the chitosan of microorganisms. When chitin deacetylation occurs, the amide I band (C = O-NHR) decreases while amide II growth occurs, indicating the prevalence of NH_2_ groups [49]. According to [50], when the range of 1500–1700 cm^−1^ is stressed, it suggests an intensification of deacetylation. The infrared spectrum chitosan *of L. hyalospora* (UCP 1266) showed the amide bands to be the most significant.

The higher DD% fungal chitosan is more appropriate for food industrial application, principally when it is used as an antimicrobial [51]. 

The viscosity of chitosan of *M. subtilissimus* (UCP 1262) (3.06 cP) was considerably higher that the of *L. hyalospora* (UCP 1266) (2.78 cP). Similar results were obtained in studies of the viscosity of fungal chitosan, where the viscosity of fungal chitosan ranged from 1.02 to 2.67 cP [52,53,54]. The commercial chitosan, extracted from natural shellfish exoskeleton, may have a deacetylation degree of 80.0–95.0% and viscosity of 20–500 cP [54].

### 2.5. Optimization of Chitosan Production by L. hyalospora (UCP 1266)

Due to the results obtained from the previous factorial design, *L. hyalospora* (UCP 1266) was selected for a further DCCR 2^2^, as it presented the highest chitosan productivity in the culture media with CSL and CWW. For this, the levels of CSL were increased and the higher levels of CWW around the central point of the previous factorial design were maintained. Since the studied concentrations may have allowed a higher activity of chitin deacetylase, the enzyme was responsible for the deacetylation of chitin in cell wall [55].

The results of the experiments that evaluated the influence of different concentrations of CSL and CWW on the production of biomass and chitosan by *L. hyalospora* (UCP 1266) are shown in Table 6.

From the calculation of the coefficients, we obtained an equation adjusted to the experimental data, with the effects of the two independent variables *x* and *y* (CSL and CWW) on the production of chitosan Equation (1).
Z_Chitosan_(mg/g) = −191.65 + 11.04x − 0.73x^2^ + 57.81y – 3.97y^2^ + 0.012xy(1)

According to the response surface graph (Figure 7) and Equation (1), the highest chitosan production by *L. hyalospora* (UCP 1266) (63.18 mg/g) was achieved in medium containing the highest concentration of CSL (+1.41) and intermediate concentration of CWW (center point—0) (assay 6). The coefficient of determination (*R^2^*) of the obtained model was 0.85, showing good suitability of the experimental data.

The analysis of these results suggests a sustainable culture medium with concentrations of cassava waste water and corn steep liquor as nutritional sources of carbon and nitrogen for optimized production of biomass and chitosan by *L. hyalospora* UCP 1266. The use of low cost substrates as nutritional sources in culture media decreases the final value of the by-products, mainly in industrial production. Microbial biomass offers economic advantages for industrial scale production on chitosan obtained from crustacean shells, as it does not require high solvent amounts and high temperatures during the extraction process, in addition to obtaining the polysaccharide in a short time.

## 3. Materials and Methods 

### 3.1. Microorganisms

*M. subtilissimus* UCP 1262 and *L. hyalospora* UCP 1266 were isolated from the soil of the city of São José do Belmonte (S 07° 51′37″ and W 038°45′35″; Pernambuco, Brazil). Fungi were identified and deposited in the Culture Collection of the Nucleus of Research in Environmental Sciences and Biotechnology, Catholic University of Pernambuco, Brazil, and registered in the World Federation Culture for Collection (WFCC). The strains were kept in Sabouraud agar medium (consisting of agar 20 g, peptone 10 g, glucose 40 g, distilled water 1000 mL, and pH adjusted to 6.0), at 5 °C.

### 3.2. Substrates

The substrates used for lipid and chitosan production were cassava waste water (CWW), kindly provided by the cassava processing plant located in the municipality of Carnaíba, Pernambuco, and corn steep liquor (CSL), obtained from Corn Products do Brasil Ltd. (Cabo de Santo Agostino, Pernambuco, Brazil). Aliquots of CSL were subjected to elemental analysis to determine the amounts of carbon, hydrogen, and nitrogen (%) using an EA 1110 analyzer (Carlo Erba Instruments). Table 1 shows the chemical composition of the cassava waste water (CWW) [23] and CSL.

### 3.3. Conditions of Culture and Biomass Production

The fungi were grown in Petri dishes (9 cm in diameter) containing Sabouraud agar medium, and were incubated at 28 °C for 5 days until sporulation. Then, the spores were transferred to Erlenmeyer flasks (250 mL) containing sterile distilled water to prepare a suspension of 10^7^ spores/mL. Aliquots (1 mL) of these spore suspensions were used as inoculum and transferred to Erlenmeyer flasks (250 mL) with 100 mL of the media containing CSL and CWW, according to the factorial design (Table 1).

The flasks were incubated at 28 °C on an orbital shaker at 150 rpm for 120 h. At the end of this period, the pH of the media was terminated using a potentiometer. The biomass was centrifuged and washed with distilled water, and mycelia fragments were removed from the biomass, stained with blue cotton dye, and observed under the optical microscope. The remaining biomass was lyophilized and kept in a desiccator to determine the constant weight by gravimetry.

### 3.4. Morphological Analysis in Scanning Electron Microscopy (SEM)

The biomass was washed in phosphate-buffered saline (PBS), pH 7.2, and fixed with glutaraldehyde 2.5% in fosfate buffer, 0.1 M, pH 7.4, for 1 h at room temperature. In the post-fixation, malachite green 0.05% was solubilized in phosphate buffer for 1 h at room temperature in dark conditions. They were then subjected to the dehydration process with ethanol in proportions of 50%, 70%, 90%, and 100%. Samples were then placed on aluminum supports and analyzed by scanning electron microscopy (LSM JEOL 5600 LV).

### 3.5. Extration Chitosan

Chitosan was extracted as described by [56]. Briefly, the biomass was deproteinized with the addition of sodium hydroxide solution (1 M, 30 mL, *v*/*v*) and subjected to 121 °C for 15 min. The insoluble alkali fraction was separated by centrifugation (4000 rpm, 15 min). The remaining biomass was washed several times with saline solution (0.85%) and cold distilled water to reach pH 7.0. The obtained residue was treated with acetic acid (2%, 30 mL, *v*/*v*) for 15 min, 100 °C; centrifuged (4000 rpm); and filtered. The supernatant was alkalized to pH 9.0, stored at 4 °C for 24 h, and centrifuged (4000 rpm, 15 min) until chitosan precipitated. The chitosan was washed with cold distilled water and saline solution at pH 7.0 and then lyophilized.

### 3.6. Characterization of Chitosan

#### 3.6.1. Infrared Spectroscopy (FTIR)

The chitosan was previously dried overnight at 60 °C under reduced pressure and homogenized with 100 mg potassium bromide (KBr). Discs prepared with potassium bromide were placed to dry for 24 h at 110 °C under reduced pressure. Infrared ray spectroscopy was performed using a Fourier transform (FTIR) BRUKER Mod. IFS. Discs with only potassium bromide were used as reference. The maximum intensity of the absorption bands was measured by the baseline.

#### 3.6.2. Deacetylation Degree

The absorbance bands 1320 cm^−1^ and 1420 cm^−1^ were used to measure the degree of acetylation, according to [57], to later obtain the deacetylation degree of chitosan, as shown in the following Equation (2):DD% = A1320/A1420 = 0.3822 + 0.03133(2)
where A1320 is absorbances of chitosan at 1320 cm^−1^, and A1420 is absorbances of chitosan 1420 cm^−1^.

#### 3.6.3. Viscosity Determination

The viscosity of 1% chitosan in 1% acetic acid solution was determined using the automatic viscometer (Brookfield, Middleboro, MA, USA; TC 500).

### 3.7. Selection of Waste Concentrations in Culture Using Factory Design

To study the effect of the independent variables (CSL and CWW) and their interaction on the response variables (biomass and chitosan), 2^2^ factorial design was performed, consisting of eight assays with four central points (Table 2).

### 3.8. Central Composite Rotational Design (CCRD) for Optimization of Chitosan Production by Lichtheimia hyalospora (UCP 1266)

To establish the optimal conditions for chitosan production by *Lichtheimia hyalospora* (UCP 1266), we performed a CCRD 2^2^ to analyze the effects of CWW (5.58–8.41%; *v*/*v*) and CSL (3.58–6.41%; *v*/*v*) (independent variables). The experimental design consisted of 12 runs that included 4 repetitions at the center point. The 12 assays were prepared in standard order. In each experiment, we calculated the biomass and chitosan production. Inoculum, incubation period, and biomass yield are described in Section 3.3.

## 4. Conclusions

The results of this study showed the potential for *Mucor subtilissimus* (UCP 1262) and *Lichtheimia hyalospora* (UCP 1266) to convert agro-industrial residues (CWW and CSL) for biomass and chitosan production. The association of CSL and CWW are promising substrates that reduce the cost to obtain effective chitosanpolymer production from Mucoralean strains. The dimorphism negatively influenced the biomass and chitosan productions by *M. subtilissimus* (UCP 1262). However, this is the first time that the effect of cassava waste water (CWW) and corn steep liquor (CSL) levels on the presence of budding cells, yeast-like cells, and arthropod short hyphae has been described. The big pellets’ diameter ranges were related to higher chitosan yields by *L. hyalospora* (UCP 1266). The investigation with Mucoralean strains showed a quality source for chitosan that was produced by *L. hyalospora* (UCP 1266), as well as the degree of deacetylation and its ease of manipulation in the laboratory.

## Figures and Tables

**Figure 1 ijms-21-04286-f001:**
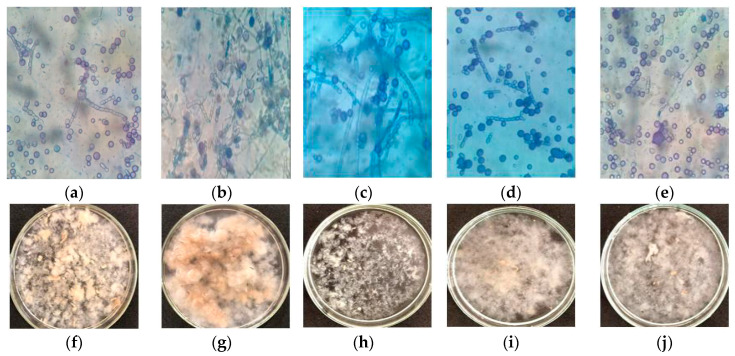
Morphological aspects of *Mucor subtilissimus* UCP 1262 cultivated in different concentrations of CSL and CWW according to 2^2^ factorial design for 120 h. (**a**,**f**): Assay 1; (**b**,**g**): assay 2; (**c**,**h**): assay 3; (**d**,**i**): assay 4; (**e**,**j**): center point. Assay 1: 2% CSL and 4% CWW; assay 2: 6% CSL and 4% CWW; assay 3: 2% CSL and 8% CWW; assay 4: 6% CSL and 8% CWW; center point: 4% CSL and 6% CWW. Bars: 10 μm.

**Figure 2 ijms-21-04286-f002:**
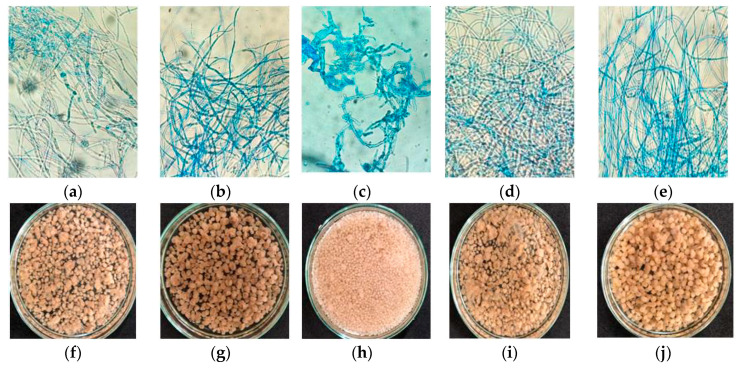
Morphological aspects of *Lichtheimia hyalospora* UCP 1266 cultivated in different concentrations of CSL and CWW according to the 2^2^ factorial design for 120 h. (**a**,**f**): Assay 1; (**b**,**g**): assay 2; (**c**,**h**): assay 3; (**d**,**i**): assay 4; (**e**,**j**): center point. Assay 1: 2% CSL and 4% CWW; assay 2: 6% CSL and 4% CWW; assay 3: 2% CSL and 8% CWW; assay 4: 6% CSL and 8% CWW; center point: 4% CSL and 6% CWW. Bars: 10 μm.

**Figure 3 ijms-21-04286-f003:**
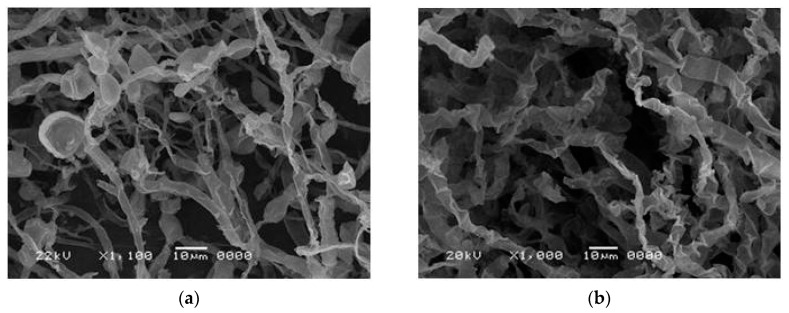
Scanning electron micrograph of mycelium of *Mucor subtilissimus* (UCP 1262) and *Lichtheimia hyalospora* (UCP 1266). (**a**) *M. subtilissimus* (UCP 1262) after cultivation in assay 2 of the factorial design (CSL 6% and CWW 4%) and (**b**) *L. hyalospora* (UCP 1266) after cultivation in assay central point of the factorial design (CSL 4% and 6% CWW).

**Figure 4 ijms-21-04286-f004:**
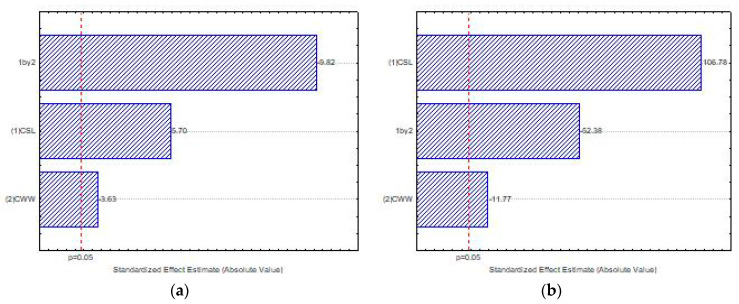
Pareto chart of standardized effects of corn steep liquor (CSL) (1) and cassava waste water (CWW) (2) on the biomass production by *Mucor subtilissimus* (UCP 1262) (**a**) and *Lichtheimia*
*hyalospora* (UCP 1266) (**b**).

**Figure 5 ijms-21-04286-f005:**
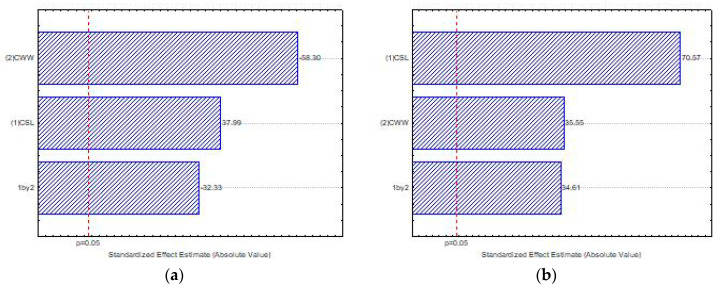
Pareto chart of standardized effects of CSL (1) and CWW (2) on the chitosan production by *Mucor subtilissimus* (UCP 1262) (**a**) and *Lichtheimia hyalospora* (UCP 1266) (**b**).

**Figure 6 ijms-21-04286-f006:**
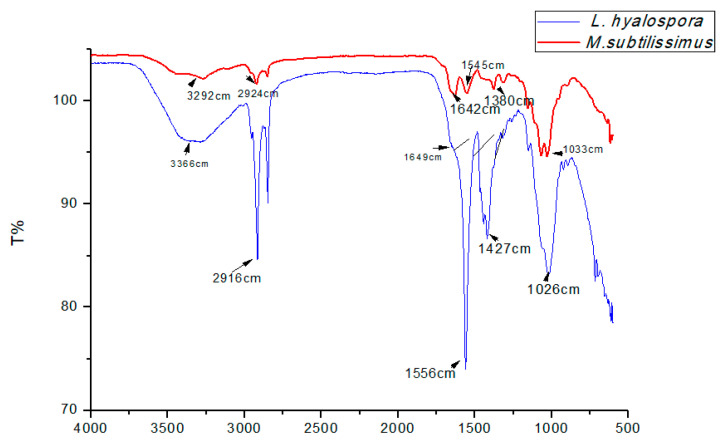
Infrared spectrum of chitosan produced by *M.subtilissimus* (UCP 1262) and *L. hyalospora* (UCP 1266).

**Figure 7 ijms-21-04286-f007:**
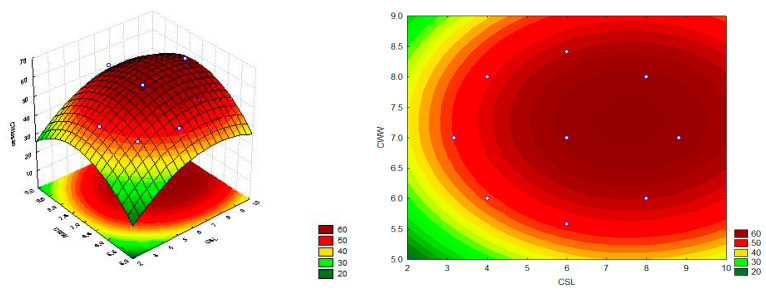
Response surface graphs of the chitosan production by *L. hyalospora* (UCP 1266) grown in medium formulated with CSL and CWW for 120 h at 28 °C.

**Table 1 ijms-21-04286-t001:** Percentage of nitrogen, carbon, oxygen, and sulfur present in the composition of cassava waste water (CWW) and corn steep liquor (CSL).

Substrate	Carbon	Nitrogen	Oxygen	Sulfur
Cassava Waste Water (CWW)	33.35	2.04	6.74	0
Corn Steep Liquor (CSL)	34.84	7.06	6.59	1.18

**Table 2 ijms-21-04286-t002:** Independents variables and levels used in the 2^2^ full-factorial design for growth of *M. subtilissimus* and *L. hyalospora.*

Variables		Levels	
	−1	0	+1
Corn Steep Liquor—CSL (%, *v*/*v*)	2	4	6
Cassava Waste Water—CWW (%, *v*/*v*)	4	6	8

Minimum level (−1); intermediate level (0); Maximum level (+1).

**Table 3 ijms-21-04286-t003:** Factorial design applied to production of biomass and chitosan by *Mucor subtilissimus* (UCP 1262) and *Lichtheimia hyalospora* (UCP 1266) using corn steep liquor (CSL) and wastewater (CWW) as alternatives substrates.

Assays	Substrates		*Mucor subtilissimus* (UCP 1262)			*Lichtheimia hyalospora* (UCP 1266)		
	CSL	CWW	pH	Biomass (g/L)	Chitosan (mg/g)	pH	Biomass (g/L)	Chitosan(mg/g)
**1**	2	4	7.1	1.725	18.87	5.8	2.675	23.31
**2**	6	4	6.2	4.832 *	32.41 *	5.2	6.540 *	29.84
**3**	2	8	6.9	2.963	13.87	5.8	3.661	23.49
**4**	6	8	6.9	2.140	14.96	5.6	4.982	42.56
**5**	4	6	6.4	4.752	19.36	5.1	6.345	44.60
**6**	4	6	6.2	4.527	19.02	5.2	6.298	44.91
**7**	4	6	6.4	4.269	18.92	5.2	6.291	44.86
**8**	4	6	6.4	4.582	19.18	5.1	6.319	45.03 *

(*) Maximum yield.

**Table 4 ijms-21-04286-t004:** Comparison of biomass and chitosan produced by *Mucor subtilissimus* UCP 1262 and *Lichtheimia hyalospora* UCP 1266 cultivated on agro-industrial substrates with the results described by the literature on Mucoralean fungi.

Fungal Strain	Medium Composition	Cultural Conditions	Biomass (g/L)	Chitosan (mg/g)	References
*Mucor subtilissimus* UCP 1262	6% CSL and 4% CWW	SmF, 28 °C, 150 rpm, 120 h	4.83	32.41	Present study
*Lichtheimia hyalospora* UCP 1266	4% CSL and 6% CWW	SmF, 28 °C, 150 rpm, 120 h	6.298	44.91	Present study
*L. hyalospora* UCP 1266	1% CSL and 25% papaya peel juice	SmF, 28 °C 150 rpm, 96 h	-	12.04	Kroll et al. [46]
*Cunninghamella elegans* UCP 0542	9.43% CSL and 42.5% papaya peel juice	SmF, 28 °C, 150 rpm, 96 h	-	37.25	Kroll et al. [46]
*C. elegans* UCP 0542	10% CWW and 4% CSL	SmF, 28 °C, 150 rpm, 72 h	5.67	57.82	Sharifia et al. [23]
*Rhizopu sarrhizus* UCP 0402	6% CSL and 13.24% honey	SmF, 28 °C, 150 rpm, 96 h	11.71	29.30	Berger et al. [32]
*Syncephalastrum racemosum* UCP 1302	8% CSL and 2% sugarcane bagasse	SSF, 28 °C, 96 h	32.0	25.0	Oliveira et al. [40]
*Mucor circinelloides* UCP 0050	Yam bean medium	SmF, 28 °C, 150 rpm, 96 h	20.7	64.00	Berger et al. [47]
*Mucor rouxii* ATCC 24905	Soybean meal	SSF, 25 °C, 144 h	-	34.40	Mondala et al. [5]
*Rhizomucor miehei* (ATCC 26282)	Sabouraud broth	SmF, 28 °C, 120 rpm, 168 h	4.1	13.67	Fai et al. [48]
*Mucor racemosus*	Sabouraud broth	SmF, 28 °C, 120 rpm, 168 h	3.8	11.72	Fai et al. [48]

SmF: submerged fermentation; SSF: solid-state fermentation.

**Table 5 ijms-21-04286-t005:** Infrared spectroscopy, degree of deacetylation (DD%), and viscosity of chitosan samples produced by *Mucor subtilissimus* (UCP 1262) and *Lichtheimia*
*hyalospora* (UCP 1266) in (CWW) and corn steep liquor (CSL).

Biopolymers	Infrared Spectroscopy	Degree of Deacetylation (DD%)	Viscosity (cP)
**Chitosan from *L. hyalospora* (UCP 1266)**	1649–1556	83.61	2.78
**Chitosan from *M. subtilissimus* (UCP 1262)**	1642–1545	80.28	3.06

**Table 6 ijms-21-04286-t006:** Central composite rotatable design applied to the production of biomass and chitosan by *L. hyalospora* (UCP 1266) using CSL and CWW as alternatives substrates.

Assays	CSL (%)	CWW (%)	pH	Biomass (g/L)	Chitosan (mg/g)	Chitosan (mg/L)
**1**	4	6	5.4	6.32	44.88	283.64
**2**	4	8	5.6	5.59	46.83	261.78
**3**	8	6	5.4	9.34	51.98	485.49
**4**	8	8	5.4	8.79	54.03	474.92
**5**	3.17	7	5.8	5.48	46.89	256.96
**6**	8.82	7	4.8	11.87	63.18	749.95
**7**	6	5.58	5.4	7.81	48.09	375.58
**8**	6	8.41	5.4	7.26	57.81	437.04
**9**	6	7	4.8	9.14	58.46	534.32
**10**	6	7	5.1	9.09	58.71	533.67
**11**	6	7	5.2	8.91	58.42	520.52
**12**	6	7	4.8	9.19	58.90	541.29

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
