# Peer review of "Biotechnological Strategies for Chitosan Production by Mucoralean Strains and Dimorphism Using Renewable Substrates"

_ijms, 2020, doi:10.3390/ijms21124286_

Round 1

Reviewer 1 Report

There are some aspects which may be intersecting for the scientific auditorium who is dealing with biopolymers. However, the papers seems to be written in hast without checking the requirements on the style of a scientific paper. The paper is worth to be published in the journal “International Journal of Molecular Science” but after very major revision only.

  • Line 2/3: „Production by“ – Check correct use of font.
  • Line 37: “association    to chitosan production” – Delete space between “association” and “to”. Line 46: “(N-acetyl glycosamine)    ,” – Delete space before the comma. Line 69: “The fungal cell wall    suggested sustainable route” – Delete space between “wall“ and “suggested”. Line 171: “Lichtheimia hyalospora    (assay central point” – Delete space between “hyalospora” and “(assay”. Line 349 and 350: “and    , Coordination” and „D.R.R    , R.F.S.A” – Delete space before comma each. Check the whole paper carefully if there are more redundant spaces which must be deleted.
  • Line 42 “In recent times, chitosan is one of the most important polymers on the market.” – The review doesn’t agree with this generalized statement of the authors – maybe this is true for biopolymers only.
  • Line 86: “2²” – Font size.
  • Line 91: Maybe it seems to be better to replace “[18]” with “Reference [18]”.
  • Line 96: The point is missing.
  • Table 3 and 4. First, the tables as a whole and their captions must be on the same page each. Second, each figures inside the tables must not be stretched to fit on the page (this highly non-scientific procedure must be generally avoided) – reformat the length-to-width ratio of the figures one-to-one. Furthermore, lettering must be improved (the same line spacing as in the text, always the same arrangement of the text, below the figures: centered text in general).
  • Line 115: “METZ and KOSSEN” – It this according to the style of the journal? Maybe “Metz and Kossen” is correct.
  • Line 117: Insert space between “1266” and “cultivated”.
  • Line 132: “[23;31,33]” – Replace semicolon with comma.
  • Line 143: “CSL. [39]” – Delete point.
  • Line 164: “R. oryzae” – Italic.
  • Line 166: Replace “Mm” with “mm”.
  • Line 173: Check abbreviation of “Fig1b” is according to the style of the journal.
  • Figure 1: Picture and caption must be on the same page.
  • Table 6: The table as a whole must be on the same page.
  • Line 198/199 and 201: “FIG.2A and B” – Must be corrected.
  • Figure 2 and 3: Lettering is too small and the pictures are too fuzzy.
  • Line 205 and 216: Replace “(A)”, “e” and “(B)” with “(a)”, “and” and “(b)”, respectively.
  • Line 206 and 201: “figure 3A” and “Figure 3b” – Must be corrected.
  • Line 224 and 226: “1cm” and “cm -1” – Correct unit must be used.
  • Line 229: “figure 4” – Capitalization.
  • Figure 4: X-axis: wave number (cm-1), y-axis: transmission (%). – What’s about “1031,97cm”?
  • Caption of Figure 4: There is no space between the caption and the text.
  • Line 251: The point is missing.
  • Table 8: Table and caption must be on the same page.
  • Equation (1): Nothing is defined – what’s about “ZChitosan”, “x” and “y”? Why commas instead points are used?
  • Figure 5: Lettering is too small and the pictures are too fuzzy.
  • Line 274: “cassava wastewater (CWW)” – Use correct font.
  • Line 279: Replace “e” with “and”.
  • Line 303: Delete one of the “)”.
  • Line 317: Define “A1320” and “A1420”.
  • Line 323: “performed.a” – Must be corrected.
  • Line 349 and 350: “(CNPq)-“and “(CAPES)-“ – Must be corrected.
  • References: Check carefully if all references are always fully according to the style of the journal.
  • Page 8 and 9 etc.: Arrangement of text, figures and tables must be highly improved in the whole paper.
  • Page numbers: There is something wrong.

Author Response

Referee 1

Comments   and   Suggestions   for  Authors

There are some aspects which  may  be intersecting for the scientific auditorium   who   is dealing with biopolymers. However, the papers seems to be written in  hast without checking there quirementson the style of a scientific paper. The paper is Worth to be published in the journal “International Journal of Molecular Science” but after very major revisiononly.

  • Line 2/3: „Production by“ – Check correct use of 

 I was done

  • Line 37: “association    to chitosan production” – Delete space between “association” and “to”.

I was done

  • Line 46: “(N-acetylglycosamine)    ,” – Delete space before the comma.

I was done

  • Line 69: “The fungal cell wall    suggested sustainable route” – Delete space between “wall“ and “suggested”.

I was done

  • Line 171: “Lichtheimia hyalospora    (assay central point” – Delete space between “hyalospora” and “(assay”.

I was done

  • Line 349 and 350: “and    , Coordination” and „D.R.R    , R.F.S.A” – Delete space before comma each. Check the whole paper carefully if there are more redundant spaces  which must be deleted.

 I was done

  • Line 42 “In recent times, chitosan is one  of  the  most  importante  polymers  on  the  ” – The review  doesn’t  agree  with  this  generalized  statement   of  the  authors – may  be  thisis  true for biopolymers only.

 I was done

  • Line 86: “2²” – Fontsize.

I was done

  • Line 91: Maybe it seems to  be  better  to  replace “[18]” with “Reference [18]”.

I was done

  • Line 96: The point ismissing.

I was done

  • Table 3 and 4. First, the tables as a whole and  their  captions must be  on  the  same page each.
  • Second, each figures inside the tables must not  best retched  to fit  on the  page (this highly non-scientific procedure must   be general lyavoided) – reformat the  length-to-wid  the  ratio of  the figures one-to-one. Furthermore, lettering must  be improved (the same line spacing as in the  text, Always the same arrangement of  the text, below  the figures: centered   text  in general).

Revised as suggested

  • Line 115: “METZ and KOSSEN” – It this according to  the  style of  the  journal? Maybe “Metz and Kossen” is correct.

Revised as suggested

  • Line 117: Insert space  between “1266” and “cultivated”.

I was done

  • Line 132: “[23;31,33]” – Replace semicolon  with 

I was done

  • Line 143: “CSL. [39]” – Delete point.

I was done

  • Line 164: “R.oryzae” – Italic.

I was done

  • Line 166: Replace “Mm” with “mm”.

I was done

  • Line 173: Check abbreviation  of “Fig1b” is  according    to  the  style  of the 

Revised as suggested

  • Figure 1: Picture andcaption must beonthesamepage.

Revised as suggested

  • Table 6: The table as a whole must be on the same 

Revised as suggested

  • Line 198/199 and 201: “FIG.2A and B” – Must be

I was done

  • Figure 2 and3: Lettering is too small  and  the  pictures are too fuzzy.

Revised as suggested

  • Line 205 and 216: Replace “(A)”, “e” and “(B)” with “(a)”, “and” and “(b)”, respectively.

I was done

  • Line 206 and 201: “figure 3A” and “Figure 3b” – Must be

I was done

  • Line 224 and 226: “1cm” and “cm -1” – Correctun it must be used.

I was done

  • Line 229: “figure 4” – Capitalization.

I was done

  • Figure 4: X-axis: wavenumber (cm-1), y-axis: transmission (%). – What’s about “1031,97cm”?

Revised as suggested

  • Captionof Figure 4: There is no space between  the  caption  and the text.

I was done

  • Line 251: The point is

I was done

  • Table 8: Table and caption must be on the same page.

I was done

  • Equation (1): Nothing is defined – what’s about “Z Chitosan”, “x” and “y”? Why commas instead points are used?

Revised as suggested

  • Figure 5: Lettering is too small and the pictures are too fuzzy.

Revised as suggested

  • Line 274: “cassava wastewater (CWW)” – Use correct font.

I was done

  • Line 279: Replace “e” with “and”.

I was done

  • Line 303: Delete one of the “)”.

I was done

  • Line 317: Define “A1320” and “A1420”.

Revised as suggested

  • Line 323: “performed.a” – Must be

I was done

  • Line 349 and 350: “(CNPq)-“and “(CAPES)-“ – Must be corrected.

Revised as suggested

  • References: Check care  fully  if  all references are Always  fully according to the style of the journal.
  • Page 8and 9 etc.: Arrangement of text, figures and  tables must be  highly  improved in the whole 

Revised as suggested

  • Page numbers: There is something

Revised as suggested

Reviewer 2 Report

 In this work, the influence of corn steep liquor (CSL) and cassava wastewater (CWW) as carbon and 24 nitrogen sources on the morphology and production of biomass and chitosan by Mucor subtilissimus 25 UCP 1262 and Lichtheimia hyalospora UCP 1266 were investigated. The abstract is meaningful and clearly describes a comprehensive summary of the research. The concept itself is interesting and they systematically designed the experiments. The manuscript is very informative, and the way information has been presented is quite impressive. All the conclusions and claims throughout the study have been well supported by the data. There are a good number of figures and graphs to represent the visual description of the data to assist the reader. I have few comments on the manuscript that need to be addressed before accepting the manuscript for publication. The introduction section is poorly written. I suggest authors to improve it and describe the motivation to conduct the research. I strongly recommend the authors to add one paragraph discussing the difference between their work and the previously performed studies in the literature. The novelty of the work should be highlighted. The discussion section also needs improvements. The discussion at molecular level should be extended. The manuscript needs in-depth discussion of all results. The editing of the manuscript need attention. Also, some typos need to be corrected.

Author Response

Referee 2

Comments  and  Suggestions for Authors

 In this work, the influence  of corn steep  liquor (CSL) and cassava wastewater (CWW) as carbon and nitrogen sources on the morphology  and  production  of  biomass  and  chitosan by Mucor  subtilissimus  UCP 1262 and Lichtheimia  hyalospora UCP 1266 were  investigated. The abstract is meaning full and clearly  describes a comprehensive  summary  of  the  research. The concept it  self  is  interesting  and  they  systematically  designed  the  experiments. The manuscript  is very  informative, and  the  way  information  has  been  presented  is quite impressive. All the  conclusions  and  claims  throughout  the  study  have  been  well  supported by the data. There are a good  number  of figures and  graph  store  presente  the visual description  of  the data to    assist  the  reader. I have few comments on the manuscript that need  to be  addressed before  accepting  the  manuscript for publication.

The introduction  section  is  poorly  written.(Revised as suggested)

I suggest  authors  to improve it and  describe  the  motivation  to  conduct  the  research. (Revised as suggested)

I strongly  recommend  the  authors  to  add  one  paragraph  discussing  the  difference  between  their  work  and  the  previously  performed  studies in the  literature (I was done).

The novelty   of  the  work  should  be  highlighted (Revised as suggested).

The discussion  section  also  needs   improvements (Revised as suggested).

The discussion  at molecular level  should  be  extended (I was done).

The manuscript  needs in-depth  discussion  of  all  results. The editing of the manuscript need  attention (Revised as suggested).

Also, some  typos   need  to  be  corrected. Revised as suggested

Round 2

Reviewer 1 Report

There are some aspects which may be intersecting for the scientific auditorium who is dealing with biopolymers. Some minor revision was done according to the comments of the reviewer. However, the corrections seems to be made in hast not considering the most important remarks. The requirements on the style of a scientific paper aren’t met. The paper is worth to be published in the journal “International Journal of Molecular Science” but still after major revision.

  • Check the whole paper carefully if there are more redundant spaces which must be deleted or insert spaces where needed. For correction you have to activate the control characters in your word processing program. Sometimes lines too must be inserted before headings.
  • Line 42 “In recent times, chitosan is one of the most important polymers on the market.” – The review doesn’t agree with this generalized statement of the authors – maybe this is true for biopolymers only. Of course the authors added some information but this is not dealing with the main subject: The most important polymers on the market are still based on polyolefines such as polyethylene and polypropylene.
  • First, the figures (Figure 1 and Figure 2) and tables (Table 3 and Table 4) as a whole and their captions must be on the same page each. Second figures (Figure 1 and Figure 2) must not be stretched to fit on the page (this highly non-scientific procedure must be generally avoided, i.e. the review will not accept the paper) – reformat the length-to-width ratio of the figures one-to-one. Furthermore, the size of lettering of the figures must be improved to be nearly the same as used in the text.
  • Line 370: “cassava wastewater (CWW)” – Use correct font.
  • Line 415/416: Correct: “A1320: absorbances of chitosan at 1340 cm−1” and “A420: absorbances of chitosan at 1420 cm−1”.
  • Arrangement of text, figures and tables must be highly improved in the whole paper.
  • Page numbers: There is something wrong. The last page is “17 of 5”, for example.

Author Response

REFEREE 1

Comments and Suggestions for Authors

There are some aspects which may be intersecting for the scientific auditorium who is dealing with biopolymers. Some minor revision was done according to the comments of the reviewer. However, the corrections seems to be made in hast not considering the most important remarks. The requirements on the style of a scientific paper aren’t met. The paper is worth to be published in the journal “International Journal of Molecular Science” but still after major revision.

  • Check the whole paper carefully if there are more redundant spaces which must be deleted or insert spaces where needed. For correction you have to activate the control characters in your word processing program. Sometimes lines too must be inserted before headings.

We have done.

  • Line 42 “In recent times, chitosan is one of the most important polymers on the market.” – The review doesn’t agree with this generalized statement of the authors – maybe this is true for biopolymers only. Of course the authors added some information but this is not dealing with the main subject: The most important polymers on the market are still based on polyolefines such as polyethylene and polypropylene.

We have done.

  • First, the figures (Figure 1 and Figure 2) and tables (Table 3 and Table 4) as a whole and their captions must be on the same page each. Second figures (Figure 1 and Figure 2) must not be stretched to fit on the page (this highly non-scientific procedure must be generally avoided, i.e. the review will not accept the paper) – reformat the length-to-width ratio of the figures one-to-one. Furthermore, the size of lettering of the figures must be improved to be nearly the same as used in the text.

We have done.

  • Line 370: “cassava wastewater (CWW)” – Use correct font.

We have done.

  • Line 415/416: Correct: “A1320: absorbances of chitosan at 1340 cm−1” and “A420: absorbances of chitosan at 1420 cm−1”.

We have done.

  • Arrangement of text, figures and tables must be highly improved in the whole paper.

We have done.

  • Page numbers: There is something wrong. The last page is “17 of 5”, for example.

We have done.

Reviewer 2 Report

The authors have addressed most of the previous comments raised by the reviewer. However, there are still minor issues related to formating. I suggest authors to proofread the manuscript before submission. There are formatting issues such as empty spaces before Figure 2, Table 3, and Table 4.

Author Response

REFEREE 2 - Second Round

Comments and Suggestions for Authors

The authors have addressed most of the previous comments raised by the reviewer. However, there are still minor issues related to formating. I suggest authors to proofread the manuscript before submission. There are formatting issues such as empty spaces before Figure 2, Table 3, and Table 4.

We have done.

Round 3

Reviewer 1 Report

The paper gives insight into a topic which may be intersecting for the scientific auditorium who is dealing with biopolymers. Now the paper was highly improved as a whole. Therefore, the paper can be accepted for publication in the journal “International Journal of Molecular Science”.